# STUDY: Socially Aware Temporally Causal Decoder Recommender Systems

## Abstract

Recommender systems are widely used to help people find items that are tailored to their interests. These interests are often influenced by social networks, making it important to use social network information effectively in recommender systems, especially for demographic groups with interests that differ from the majority. This paper introduces STUDY, a Socially-aware Temporally caUsal Decoder recommender sYstem. The STUDY architecture is significantly more efficient to learn and train than existing methods and performs joint inference over socially-connected groups in a single forward pass of a modified transformer decoder network. We demonstrate the benefits of STUDY in the recommendation of books for students who have dyslexia or are struggling readers. Students with dyslexia often have difficulty engaging with reading material, making it critical to recommend books that are tailored to their interests. We worked with our non-profit partner Learning Ally to evaluate STUDY on a dataset of struggling readers. STUDY was able to generate recommendations that more accurately predicted student engagement, when compared with existing methods.

## 1 Introduction

Recommender systems are one of the major applications of AI systems and are an essential driver of many of our online experiences today. With applications ranging from e-commerce (1) and advertising platforms (2) to video platforms (3), we are relying on recommender systems to surface relevant and interesting content to individual users. In this work, we focus on recommender systems deployed in the educational setting (4) to suggest relevant literature for students in grades 1 through 12, specifically audiobook content on a platform designed to assist students with reading difficulties such as dyslexia. Prompting content that is more likely to engage the student is theorized to lead to more successful learning outcomes.

In the applied educational setting, systems can be targeted towards either teachers (5) or students (6) to suggest content, and in both cases the goal of these systems is to surface relevant and engaging educational material that is beneficial to the students' learning. Student-facing educational recommender system is built from data relevant to students interaction. with the platform, which falls into the following categories (7):

- Data about the students, or "user data"
- Data about the content to be recommended, or "item data"
- Data about the context of the current session (e.g. time of day, session device, etc.), or "context data".

Two widely used types of recommender systems are "Batch" and "Sequential". Batch recommender systems operate on representations of previous interactions, and don't model time or relative order. They include collaborative filtering based methods (8) and Embarrassingly Shallow Autoencoders (EASE) (9). Sequential recommender (10) systems operate on representations of historical user interaction as sequences (11).

The classroom setting enables socially-structured recommendation because of the availability of a clearly-defined hierarchical network, which groups students into classrooms, year cohorts, and

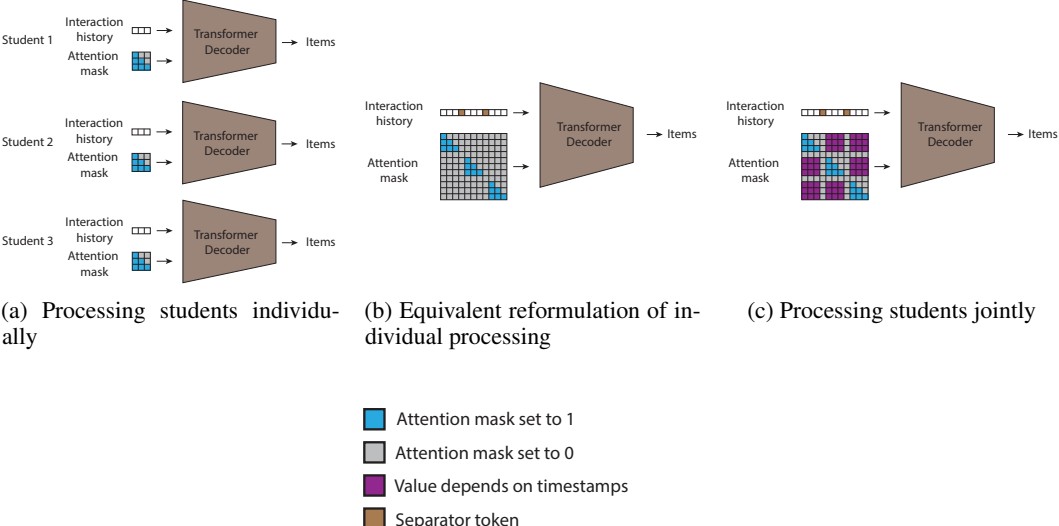

(a) Processing students individually

(b) Equivalent reformulation of individual processing

(c) Processing students jointly

Figure 1: (a) a sequential autoregressive transformer with causal attention that processes each user individually, (b) an equivalent joint forward pass that results in the same computation as (a), c) shows that introducing new nonzero values (shown in purple) to the attention mask allows information to flow across users. Predictions condition on all interactions with an earlier timestamp, irrespective of whether the interaction came from the same user or not.

schools. This makes the utilization of social recommendation systems (12) particularly attractive where the relationships between users are leveraged to make recommendations.

In this work we present Socially-aware Temporally caUsal Decoder recommender sYstems (STUDY), a sequence-based recommender system for educational content that makes use of known student hierarchies to improve recommendations. Our method does joint inference over groups of students who are adjacent in the social network. Our method utilizes a single transformer decoder network to model both within-user and cross-user interactions. This paper is organized as follows: we provide a review of the related work in Section 2, we review related previous recommender systems in Section 3, we introduce our new socially-aware recommender system in Section 4, and we present our experimental results in Section 5.3.1 before concluding in section 6.

In summary, the contributions of this paper are:

- Proposing a novel architecture and data pipeline for performing socially-aware sequential recommendation.
- Comparing the new method to modern and classical recommendation baselines.
- Performing ablations and performance breakdowns to better understand the new model.

## 2 RELATED WORK

### 2.1 EDUCATION

Recommender systems for educational content have been studied in the context of online learning programs/massive open online courses (MOOCs) (13) but are not as common for primary and secondary school student applications. Experiments with recommender systems in education have generally been limited by the lack of publicly-available large data sources - one review found only 5 experimental studies with sample sizes of over 1000 participants (14). However, versions of recommender systems have been applied in educational measurement and assessment for over four decades through computerized adaptive testing (15), where the test items presented to the test-takers depend on the current estimate of the student's ability.

In more recent literature, expansions on methods for computerized adaptive testing have been proposed for recommending new content in an adaptive learning framework (16) where content can be automatically presented to students given their particular stage of learning. These adaptive learning systems require access to some measures of student subject-matter performance and do not account for the student's interest in the material nor the social dynamics that may lead to greater engagement. Alternative approaches are needed in content recommendation contexts where a student's reading level cannot be measured or measures are not included in the available data for recommendations. In the case of our applied scenario, we do not assume that student or "user data" includes measures of student performance.

Previous studies have shown that higher levels of student motivation predicts growth in reading comprehension (17); thus promoting content that is most likely to align with a student's interests is hypothesized to produce better reading and literacy outcomes, particularly for students with reading difficulties. In the United States, often the reading content assigned in by teachers aligns to a state or district-level curriculum for a particular grade level, but for assigning reading materials outside the required texts, other strategies are needed, and we hypothesize that incorporating social connections can be an effective strategy for recommending content successfully.

In one study of a book recommendation system, using an app function that allowed users to view the reading histories of peers had a beneficial long-term effect on reading choices (18), indicating that incorporating social dynamics into recommendations may lead to more effective recommendations. In another social network analysis of second and third graders in the US (19), the researchers found that on average students were able to effectively identify peers with higher reading skills to ask these peers for help, thus even for younger learners, peer relationships may be relevant for content selected. In rural areas which sometimes lack resources to help with struggling students, a study (20) found that adolescent reading choices were often motivated by conversations and materials borrowed from friends and family, suggesting that a recommender system that includes peer preferences could also be effective for reaching the rural student population.

## 2.2 CLICK-THROUGH RATE PREDICTION

One of the popular approaches for recommender systems is click-through rate prediction (21), where the probability of a user clicking on a specific presented item is predicted. These probabilities are then used as a proxy for user preferences. Click-through Rate (CTR) models typically make predictions for a suggested next item for a user based on the user's sequence of previous interactions, user data and context data. Model architectures used in this problem range from standard models like Transformers used in Behavior Sequence Transformers (BST) (22) and Convolutional Neural Networks used in (23) to more task specific architectures such as Wide & Deep models (24) and Field-Leveraged Embedding Networks (FLEN) (25). This approach contrasts with other approaches such as neural collaborative filtering (26) and K-Nearest Neighbors (KNN) recommenders (27) where there is no attempt to explicitly model the likelihood of the user interacting with a specific item.

## 2.3 SOCIALLY-AWARE RECOMMENDATION SYSTEMS

When social connectivity information exists for users, there are many modeling approaches that leverage this information. Methods such as TrustMF (28) and Sorec (29) project user preference vectors into a latent space using matrix factorization approaches. The underlying assumption of these systems is homophily i.e. that users who are more socially connected are more likely to have similar preferences.

Deep-learning based methods have leveraged graph neural networks to learn using social connectivity structure. Methods such as DiffNet (30) and KCGN (31) utilize graph convolutional neural networks whereas methods such as GraphRec (32) and Social Attentional Memory Networks (SAMN) (33) employ graph attention mechanisms. Other notable work includes Disentangled Graph Neural Networks (DGNN) which have the capability to model non-heterogeneous relationships and utilize a memory augmented graph network (12).

In this work we take a different approach to that of previous work, which has used graph neural networks or other custom architectures with separate components to handle cross-user interactions. We utilize a single transformer decoder with a specifically-designed attention mask to do joint inference

over groups of users. With this approach we have developed a single consistent way to handle both within-user and cross-user interactions in a computationally efficient manner.

# 3 REVIEW OF BASELINE WORK

We review related work, used as baselines for comparison in the experiments section 5.3.1.

## 3.1 ITEM-BASED KNN RECOMMENDER

KNN recommender systems (34) compute the cosine similarity between the user's current feature vector and each entry in the training dataset. They then recommend to the user the $k$ distinct items with highest cosine similarity to the user's current feature vector. When feature vectors are sparse most entries in the training dataset will have a cosine similarity of exactly zero with the user's current feature vector. In this KNN implementation, we iterate over every sequence in the training dataset and featurize each item by computing a feature vector from the $h$ interactions preceding it.

## 3.2 INDIVIDUAL

Following the click-through rate prediction (35) method of recommendation this methodology takes the next-click prediction approach to recommendation, and hence treats making recommendations as a causal sequence-modeling problem. In particular, this modeling framework borrows from the language modeling literature (36) due to a similar problem setup. Concretely, given a set of students $s_j \in S$, and a set of historical item interactions $\{i_j^k : \forall j | s_j \in S, \forall k < \|s_j\|, k \in \mathbb{N}\}$ we learn a propensity function $P(i_j^k) = f(i_j^k | i_j^{k' < k}; \theta)$, where the propensity of an item at a point in time the likelihood the student will interact with that item at that time. To this end we modeled $f$ as a causal decoder-only transformer with a next-token prediction objective to maximize the following likelihood of our data $D$, $\mathbb{L}(D) = \sum_{s_j^i \in D} \log f(s_j^i | s_{j' < j}^i; \theta)$. This is the formulation we used for the model referred to as **Individual**, since inference is carried out for each individual student separately.

## 3.3 SAMN

Social Attentional Memory Networks (SAMN) (33) are a class of batch recommenders that take social network information into account. Given a a set of users, a set items and a social connectivity graph, we learn fixed embeddings for each item and each user. To score the likelihood of the user interacting a with a specific item we compute a score using the user's embedding, the item's embedding and the embeddings of known 1st degree social contacts (friends). As it is a batch recommender and does not take the sequence of interactions into account at training or test time. Given that we need to make recommendations for new users at test time we utilize the Heater framework (37) to learn default user embeddings computed from user features (in this case grade level).

## 3.4 SR-GNN

Session-based Recommendation with Graph Neural Networks (SR-GNN) (38) are a class of sequential recommenders that utilise graph neural networks. Concretely they take the sequence of items interacted with in the current session and construct an item graph, with two items $A$ and $B$ being connected if the user interacted with item $B$ directly after interacting with item $B$. A gated graph neural network is then used to compute a session embedding from the input graph of items. This session embedding is used to rank items to produce recommendations.

# 4 METHOD

We present our new Socially-aware Temporally Causal Decoder Recommender System (STUDY), enabling the efficient use of the unique social structure inherent to students in schools.

## 4.1 STUDY

We motivate our model by observing that for students with few previous interactions, we can rely on data from other students with similar preferences to seed the model to improve predictive performance. Concretely, we concatenate the interaction sequences of multiple students within the same classroom. This precludes using a causal sequence modeling approach to model this problem, since some item-interactions for students presented earlier in the sequence could have occurred at a later point in time relative to item-interactions for the students presented later in the sequence. Modeling data represented in this format using causal sequence modeling would lead to anti-causal data leakage and the model would learn to make recommendations conditioned on information not available at inference time.

Hence we introduce temporally causal masking into our model: a change to our model's forward pass using a training process similar to causal sequence modeling that respects the causal relationships in our data as shown in Figure 1. Conceptually we concatenate the user vectors of students in the same classroom and allow predictions for a particular item to condition on all interactions that happened in the past, both within-user and cross-user. In more detail, if there is a subset of users $u^1, u^2, \cdots, u^n$ who are all in the same classroom, with interaction sequences $S^1, S^2, \cdots, S^n$, and with timestamp vectors $T^1, T^2, \cdots T^2$ where $t_j^i$ is the timestamp of the interaction described at $s_j^i$ - and each user vector $S^n$ and timestamp vector $T^n$ is terminated with a separator token - we define the concatenated classroom vectors generated by the procedure described in Section 5.2 as

$$\hat{S} = \left( S^1 S^2 \cdots S^n \right)$$

$$\hat{T} = \left( T^1 T^2 \cdots T^n \right)$$

We define the matrix $M$

$$m_{i,j} = \mathbb{1}_{\hat{t}_i < \hat{t}_j}$$

as the temporally causal mask matrix. This matrix is used as the mask in our attention operator instead of the usual causal mask used in decoder-only transformers. Hence our we redefine the attention operator in the decoder-only transformer as follows.

$$A = \text{Softmax}\left(\frac{QK^T}{\sqrt{d_k}}\right) \odot M$$

$$\text{Attention}(Q, K, V) = AV$$

where $Q$ is the query matrix, $K$ is the key matrix and $V$ is the value matrix. With this modification we can use next-item prediction sequence modeling to train the model without anti-causal information leakage, utilizing a multihead generalization of this attention mechanism (39). We call the model defined by this procedure **STUDY**.

## 5 EXPERIMENTS

### 5.1 DATA

We test STUDY on a dataset of historical interactions with an educational platform collected by our nonprofit partner, Learning Ally. This platform recommends and provides access to audiobooks with the goal of promoting reading in students with dyslexia. The data offered was anonymized, with each student, school and district identified only by a unique randomly generated identification number. Furthermore, all descriptive student data was only available as highly aggregated summaries. There are historical records of interactions between students and audiobooks in the dataset. For each interaction recorded we have a timestamp, an item ID and an anonymized student ID, an anonymized school ID and a grade level. This data was collected over two consecutive school years containing over 5 million interactions per each school year totalling over 10 million interactions. These interactions come from a cohort of over 390,000 students. We use the data from the first school year as our training dataset and split the data from our second school year into a validation dataset and a test dataset. This split was done according to the temporal global splitting strategy (40). This was done to model the scenario of deployment as realistically as possible. To partition the data

| Evaluation Subset | $n$ | KNN(%) | SAMN(%) | SRGNN (%) | Individual (%) | STUDY(%) |
|---|---|---|---|---|---|---|
| All | 1 | $16.10 \pm 0.11$ | $0.32 \pm 0.02$ | $31.46 \pm 0.15$ | $31.78 \pm 0.15$ | $\mathbf{32.21 \pm 0.15}$ |
| | 3 | $23.95 \pm 0.15$ | $2.01 \pm 0.05$ | $37.41 \pm 0.16$ | $37.73 \pm 0.16$ | $\mathbf{38.09 \pm 0.15}$ |
| | 5 | $27.38 \pm 0.15$ | $3.64 \pm 0.09$ | $39.38 \pm 0.17$ | $39.89 \pm 0.15$ | $\mathbf{40.26 \pm 0.16}$ |
| | 10 | $31.70 \pm 0.15$ | $6.87 \pm 0.10$ | $42.68 \pm 0.19$ | $43.42 \pm 0.19$ | $\mathbf{43.79 \pm 0.22}$ |
| | 20 | $35.15 \pm 0.15$ | $11.56 \pm 0.15$ | $46.24 \pm 0.19$ | $46.71 \pm 0.18$ | $\mathbf{47.86 \pm 0.16}$ |
| Non-continuation | 1 | $2.80 \pm 0.04$ | $0.33 \pm 0.02$ | $3.55 \pm 0.04$ | $4.01 \pm 0.04$ | $\mathbf{4.05 \pm 0.05}$ |
| | 3 | $5.31 \pm 0.07$ | $1.95 \pm 0.04$ | $11.49 \pm 0.10$ | $12.27 \pm 0.10$ | $\mathbf{12.58 \pm 0.10}$ |
| | 5 | $6.55 \pm 0.06$ | $3.56 \pm 0.07$ | $14.31 \pm 0.09$ | $15.44 \pm 0.10$ | $\mathbf{15.86 \pm 0.10}$ |
| | 10 | $9.26 \pm 0.09$ | $6.68 \pm 0.09$ | $19.17 \pm 0.14$ | $20.71 \pm 0.15$ | $\mathbf{21.08 \pm 0.12}$ |
| | 20 | $12.22 \pm 0.11$ | $11.29 \pm 0.15$ | $24.91 \pm 0.15$ | $25.76 \pm 0.14$ | $\mathbf{27.60 \pm 0.18}$ |
| Novel | 1 | $0.32 \pm 0.02$ | $0.32 \pm 0.02$ | $0.47 \pm 0.03$ | $0.55 \pm 0.03$ | $\mathbf{0.73 \pm 0.03}$ |
| | 3 | $1.72 \pm 0.04$ | $1.87 \pm 0.05$ | $4.23 \pm 0.07$ | $4.05 \pm 0.06$ | $\mathbf{4.90 \pm 0.09}$ |
| | 5 | $2.56 \pm 0.07$ | $3.45 \pm 0.06$ | $6.44 \pm 0.07$ | $6.65 \pm 0.07$ | $\mathbf{7.75 \pm 0.10}$ |
| | 10 | $5.03 \pm 0.05$ | $6.47 \pm 0.12$ | $11.22 \pm 0.13$ | $12.00 \pm 0.12$ | $\mathbf{12.96 \pm 0.11}$ |
| | 20 | $7.85 \pm 0.09$ | $10.99 \pm 0.11$ | $17.41 \pm 0.14$ | $17.65 \pm 0.12$ | $\mathbf{20.03 \pm 0.15}$ |

Table 1: Hits@n percentage metrics for the different recommendation models evaluated on the historical data in the test split, across three subsets: *all*, *non-continuation* and *novel*.

from the second school year into a test set and a validation set we split by student, following the user split strategy (40). If a data split does not contain at least a full academic year then the distributions would not match due to seasonal trends in the data.

Overall this dataset is well suited to studying social recommendation algorithms due to the existence of implied social connections through known proximity and also due to the large amount of interaction data on record. The existing book selections were made through either student choice or teacher recommendation, where often the teacher-assigned content aligned to materials assigned to the whole class or required curriculum. Interactions with the assigned content, however, were still up to the learner, and thus we believe the existing data is a good fit for modeling preferences and likely engagement with content. Further details on the data including summary statistics can be found in Appendix A

## 5.2 PREPROCESSING

In order to get the training data representation, we express the items as tokens. The top $v$ most popular items get a unique and sequential integer as their token, while the rest of the items get assigned to an out-of-vocabulary token. The student interaction history will therefore become a list of such tokens associated with a time point.

Additional processing steps are then taken based on the model type used downstream. For transformer models: we split the student history into slices based on a context window of length $c$. For models that process students jointly: we split the sequence associated with each student into segments of length $s$, $s < c$, then compose sequences of length $c$ by joining segments from *multiple* students in the same classroom, taking care to use a separator token.

Additional information, including a diagram of our preprocessing steps, are in Appendix B.

## 5.3 EVALUATING MODELS

We implement **KNN**, **SR-GNN**(38) and **SAMN**[1] (22) as baseline models, a transformer-based model that does inference for each student separately, which we will call **Individual**, as well as a transformer that operates over groups of students called **STUDY**.

We compare results from the Individual model, STUDY model, the item-based KNN baseline, SR-GNN (38) as a recent baseline and SAMN (33) as a social baseline . We tuned the hyperparameters learning rate on the validation set and report final results on the test set. We took the both the context length $c$ and the segment size $s$ for our transformer models to be 65, enough to the full history of most students in our dataset. Details about further hyperparameters and compute can be found in

---

[1]We used the author's repository `https://github.com/chenchongthu/SAMN` as a guideline. We found a discrepancy between this code and the method described in the paper, but it didn't affect final performance.

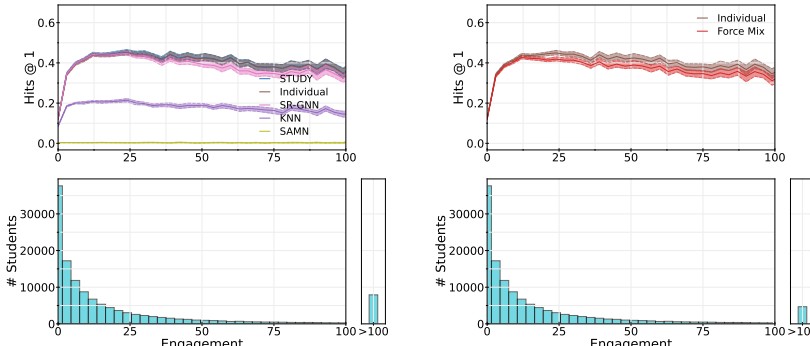

Figure 2: Models broken down by student engagement, accompanied by a histogram of student engagement in the lower chart. (left) Hits@1 across five models KNN, SAMN, SRGNN, Individual and STUDY. SRGNN matches STUDY and Individual until sequences of length 50, and underperforms on longer sequences (right) A comparison of the Individual against the Force Mix ablation.

Appendix C. Hits@n scores was used as our evaluation metric, where hits@n is the percentage of interactions when the actual item interacted with falls within the top $n$ recommendations from the model under evaluation. Since we observe that students tend to repeatedly interact with an item multiple times before completing it, we additionally evaluate our models on the subset of the dataset where the student is interacting with a different item to the item previously interacted with, referred to as *non-continuation* evaluation. We also evaluate on the subset of the dataset where the students are interacting with an item for the first time, referred to as *novel* evaluation. This motivated by the fact that we are interested in the power of our recommendation systems to engage students with new items in order to maximize time spent on the educational platform. Aggregate statistics are computed per student then averaged over students to prevent students with large numbers of interactions from dominating the reported statistics. We also examine the relevant performance of these models on different slices of data, looking at co-factors such as demographic characteristics and school performance. We present the results of this experiment in section 5.3.1.

### 5.3.1 RESULTS AND ANALYSIS

Table 1 shows the performance of the models STUDY, Individual, KNN, SR-GNN and SAMN on the test split of audiobook usage data. Uncertainties shown are always $95\%$ confidence intervals computed over 50 bootstraps. We observe that both transformer models, Individual and STUDY, as well as the GNN model SR-GNN, largely outperform KNN and SAMN, with the STUDY model outperforming the Individual model. We see that the social model SAMN, derived from the collaborative filtering family of models, fails to pick up on the sequential patterns in the dataset, such as users revisiting the exact same item or similar items. This is exhibited by SAMN having similar performance in the evaluation subsets *all*, *non-continuation* and *novel*. The performance differences are most pronounced when evaluated on the entire test set as seen in the *all* section of the table, but also holds up when evaluated across the more difficult *non-continuation* and *novel* test subsets. Crucially, with the STUDY model outperforming the individual model, we can see that leveraging the social hierarchy of our users to do joint predictions leads to improved recommendations.

In Table 3 we see the relative performance of the models under examination to be constant, with STUDY outperforming Individual. Individual slightly outperformed SR-GNN, which in turn outperforms KNN. SAMN trailed behind with almost 0 hits@1, we attribute this to SAMN's non-sequential nature. This ordering is the same when slicing by demographic variables such as metro-code (which describes schools as being in urban, suburban, rural or town areas), school socio-economic indicators which indicate the level of wealth of the area the in the vicinity of a school. We also observe the same ordering of models by performance when slicing by academic variables such as classroom reading scores. In Figure 2 we slice model performance by student engagement, which we measure by the number of interactions the student has on record. Here we see similar rela-

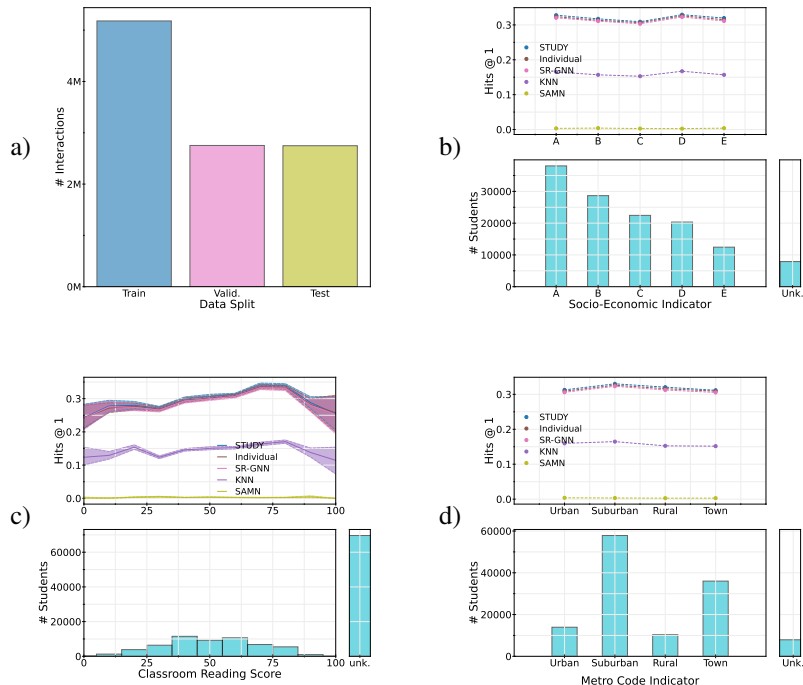

Figure 3: a) the size of train, validation and test data splits given by the number of student-title interactions recorded. Y-axis is millions of interactions. b), c) and d), hits@1 and histograms of number of students across slices (b) socio-economic indicator, (c) classroom reading score and (d) Metro code which describes schools as being Urban, Suburban, Rural or Town.

tive performance between STUDY, Individual and SR-GNN for students with up to 50 interactions, but for students with more engagement, we see SR-GNN starting to underperfrom its transformer counterparts, with KNN and SAMN lagging behind across all values of engagement.

## 5.4 ABLATION EXPERIMENTS

**Force Mix**: In our model because we set segment size $s$ equal to context size $c$ we only do joint inference over groups of students when processing a student who does not have enough previous interactions to fill the transformer's context. We experiment with shorter segment size $s = 20 \ll c$ as per the definitions in Section 5.2. Practically, this leads to the model always sharing its context between students in a classroom when possible, even for students have enough history to fill the transformer context. In Figure 2 The model from the Force Mix ablation only matches Individual on students who have up to about 17 interactions on the platform, and starts to underperform on students with more interactions. Given that the segment length for the Force Mix model is 20, it is at students with 20 previous interactions where Force Mix starts to forgo the available history for the student at hand in favor of conditioning on data from other peer students. From here we can conclude that conditioning on peer student history is beneficial if it is done in addition to conditioning on all available history for a student, but not if it comes at the cost of conditioning on less history than available for the particular student.

**Classroom Grouping**: In STUDY we do joint inference over students in the same classroom. We ablate the importance of using this particular grouping. Concretely, we experiment with grouping students who are in the same district and school year as being in a single group. We also experiment with grouping all students in the dataset into a single group, which results in completely random groups of students being jointly processed together. In Figure 4(left) we compare the performance of our model that uses classrooms to group students for joint inference compared to a model that uses intersection of district and school year to group students, to a model that uses a single group as well

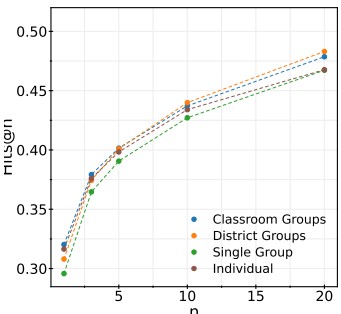 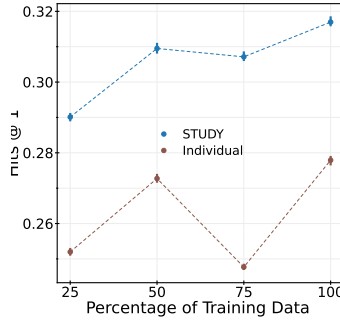

Figure 4: Figure (left): grouping the students for STUDY by classroom, grouping by the intersection of district and school year, grouping randomly, and doing individual inference. Figure (right): training with subsets of the training data of various sizes on Individual and STUDY.

as to a model that does inference for each student individually. We can see that using classrooms for grouping results in the best performance, that using the intersection of district and school year for grouping performs slightly worse, and that putting all students in the same group performs similarly to individual processing. From here we can conclude that using groups of users whom we expect to have correlated interests is necessary for the performance of our model and using poorly designed groups can harm model performance.

**Data Tapering**:We compare the effect of using only a subset of the available data and compare the performance of STUDY and Individual. We compare the use of 25%, 50%, 75% and the entire dataset, with the aim of discerning the effect of using social information on the data efficiency of the system. In Figure 4(right) we see that STUDY outperforms the Individual recommender across all data subsets used to train the models, confirming the benefit of adopting social recommender systems such as STUDY even when in a data-constrained environment. We also note that both models witness a performance drop when the amount of data used increases from 50% to 75%, suggesting that not all additional data is beneficial.

## 6 CONCLUSION

In this paper we present STUDY, a socially aware recommendation system that leverages cross-user information at inference time and we demonstrate its applicability to the practical problem of book recommendation for children inside a classroom. This is an important problem, as engagement with reading materials from an early age can positively impact language acquisition, communication skills, social skills and literacy skills.

Our novel STUDY method uses attention masks that are causal with respect to interaction timestamps and is able to process both within-user and across-user interactions using a single forward pass through a modified transformer decoder network. It avoids complex architectures and circumvents the need for graph neural networks which are notoriously difficult to train; thus, STUDY is an efficient system that can be deployed by partners with limited computational bandwidth that doesn't sacrifice model performance. We also compare STUDY to a number of baselines, both sequential and non-sequential, and social and non-social. We show that STUDY outperforms alternative sequential and social methods, in a variety of scenarios, as demonstrated in ablation studies.

**Limitations**: Evaluations were limited to offline evaluations on historical data, inline with much of the literature. However, these evaluations cannot account for the utility of recommended items that the user has never interacted with in the past, but would have enjoyed. Furthermore, our method is limited to structures where all the known relationships between users are homogeneous - each student in the classroom is assumed to have the same relationship with each other. Given that social dynamics in reality are more complicated, in future work we wish to explore extending this method to social networks with richer heterogeneous relationships between users where proximity between users can vary within a classroom.

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
