# APPENDIX

## A  DATA

Here we provide some descriptive statistics about the data this research was conducted on. The data was collected over two school years by our data partner Learning Ally. To protect the privacy of individuals and entities all personally identifiable information was removed from the dataset by our partner before we had access to the data. Furthermore much of the available metadata was redacted or had its granularity reduced so individuals and entities cannot be identified. All identification numbers in the dataset supplied were randomly generated and are not traceable back to individual students by us. Table 2 shows descriptive statistics about the platform interactions recorded as well as metadata describing the students under observation.

| Grade Level | Number of Students |
|---|---|
| Grade 1 | 1726 |
| Grade 2 | 6386 |
| Grade 3 | 16668 |
| Grade 4 | 28762 |
| Grade 5 | 41070 |
| Grade 6 | 49805 |
| Grade 7 | 59836 |
| Grade 8 | 57413 |
| Grade 9 | 51217 |
| Grade 10 | 31611 |
| Grade 11 | 18353 |
| Grade 12 | 14717 |
| Other | 137671 |

(a) A breakdown of the students in the dataset by grade level.

| Percentile | Time Spent on Platform |
|---|---|
| $5^{th}$ | 3m |
| $25^{th}$ | 35m |
| $50^{th}$ | 2h 35m |
| $75^{th}$ | 8h 12m |
| $95^{th}$ | 33h 33m |

(c) Percentiles of the amount of interaction time logged by students in the first data collection year.

| Percentile | Time Spent on Platform |
|---|---|
| $5^{th}$ | 2m |
| $25^{inth}$ | 29m |
| $50^{th}$ | 2h 14m |
| $75^{th}$ | 7h 34m |
| $95^{th}$ | 35h 25m |

(d) Percentiles of the amount of interaction time logged by students in the second data collection year.

| Wealth Indicator | Number of Students |
|---|---|
| A | 120281 |
| B | 91325 |
| C | 74855 |
| D | 61685 |
| E | 39083 |
| Unknown | 12108 |

(b) A breakdown of the students in the dataset by the wealth indicator associated with their school (A is highest).

| Percentile | Interaction Count |
|---|---|
| $5^{th}$ | 1 |
| $25^{th}$ | 3 |
| $50^{th}$ | 10 |
| $75^{th}$ | 29 |
| $95^{th}$ | 105 |

(e) Percentiles of the total number of interactions logged per student in the dataset after aggregating over both school years.

Table 2: This table shows some descriptive statistics relating to the students in the dataset as well as the amount data logged for each student

To generate train, validation and test splits the following procedure was followed. The data from the first school year was taken as our training dataset and we split the data from the second school year into a validation and test datasets. This first split was done according to the temporal global splitting strategy (40). This was done to model the scenario of deployment as realistically as possible. To partition the data from the second school year into a test set and a validation set we split by student, with all interactions recorded for a particular student collected in the second school year are either all the testing dataset or all the validation dataset. This second split followed the user split strategy (40)

|  | Train | Validation | Test |
|---|---|---|---|
| Number of Interactions | 5,179,466 | 2,752,671 | 2,747,699 |
| Number of Students | 237,253 | 126,049 | 126,050 |
| Number of Classrooms | 40,522 | 30,243 | 30,400 |
| Number of Districts | 2,510 | 2,378 | 2,387 |

Table 3: This table provides descriptive statistics with regards to the number of interactions we have on record for each student. Students, classrooms and districts whose interactions appear in the training set might also have interactions collected at a later date that appear in the validation and test splits as the data split was partially temporal and not purely user based

because if a data split does not contain at least a full academic year then the distributions would not match due to seasonal trends in the data. Table 3 shows some key statistics for the data splits

## B PREPROCESSING

We apply the following preprocessing steps to our data before using it to train our models. First, we represent the items under recommendation with tokens. The tokens are sequential integers for the $v$ most popular items. We take the vocabulary size $v$ to be 2000. We then assign all the remainder of the items to a single unique token used to represent out-of-vocabulary items.

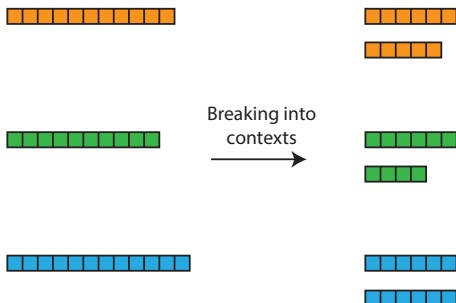

(a) For *Individual* sequences relating to different students are split into contexts of length $c$. Each context is fed into the transformer as a separate datapoint.

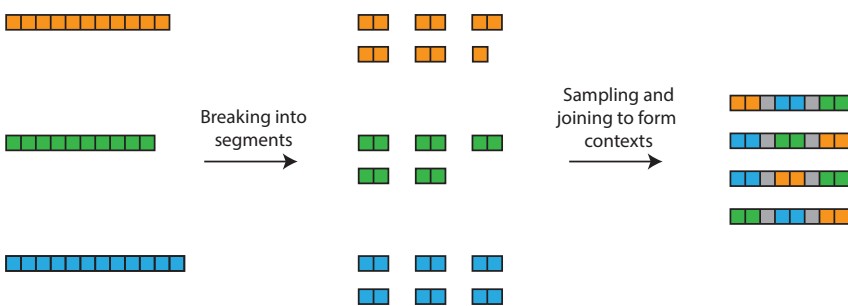

(b) For *STUDY* we split sequences relating to students into segments of length $s$, $s \leq c$. We then sample multiple chunks from different students in the same classroom. The sampled segments are then concatenated with separator tokens (shown in gray) in between them to form datapoints of at most length $c$.

Figure 5: This figure details the preprocessing pipelines used for the two transformer models, with the pipeline for *Individual* shown in (a) and the pipeline for *STUDY* shown in (b)

Additional preprocessing is applied for Individual and STUDY. For the Individual model we set a context window length $c$ and split students with interaction histories longer than $c$ items into separate data-points of at most length $c$. We took context length $c = 65$ for the transformer models. This is necessary as although the vast majority of sequences in each data split (over 92%) are under 65 entries in length, there exists a long tail of sequences with very long length.

For STUDY slightly different processing is required. We first split the sequence associated with each student into *segments* of at most length $s$, $s \leq c$. Then we compose data-points of at most length $c$ by concatenating together multiple *segments* from multiple students in the same classroom separated with separator tokens. To compose a single data-point we sample multiple segments from students in the same classroom, while satisfying the constraints that overall length of each data-point is at most $c$ and that we do not sample more than one segment from a single student. In our final model we took $s = c = 65$ Figure 5 shows a diagram explaining these procedures.

## C    EXPERIMENTAL DETAILS

In the following section we detail the choices of hyperparameters and the computational resources needed. For the KNN recommender we took K to be equal to 2. There were no further hyperparameters for the KNN recommender. The KNN recommender system had no training requirements, only preprocessing and inference, which we were able to run on the entire test split on a single machine with no GPU or TPU accelerators within a few hours. For STUDY, Individual and SAMN we used the Adam Optimizer with the following learning rate schedule

$$\alpha(s) = \begin{cases} \alpha_p * s/W & 0 < s \leq W \\ \frac{\alpha_p}{\sqrt{s-W}} & s > W \end{cases}$$

| Individual | |
|---|---|
| Peak Learning Rate $\alpha_p$ | 0.1024 |
| Warm up steps $W$ | 1000 |
| Total steps | 3500 |
| Batch Size | 131,072 |
| Number of TPUs | 32 |
| Run time | ~3 hours |

| SAMN | |
|---|---|
| Peak Learning Rate $\alpha_p$ | 0.0128 |
| Warm up steps $W$ | 350 |
| Total steps | 3500 |
| Batch Size | 524,288 |
| Number of TPUs | 32 |
| Run time | ~3 hours |

| STUDY | |
|---|---|
| Peak Learning Rate $\alpha_p$ | 0.1024 |
| Warm up steps $W$ | 1000 |
| Total steps | 3500 |
| Batch Size | 131,072 |
| Number of TPUs | 32 |
| Run time | ~3 hours |

| SRGNN | |
|---|---|
| Peak Learning Rate $\alpha_p$ | 0.0001 |
| Warm up steps $W$ | 1000 |
| Total steps | 3500 |
| Batch Size | 131,072 |
| Number of TPUs | 32 |
| Run time | ~7 hours |

Table 4: This table shows hyperparameter values as well as computation resources for STUDY, Individual and SAMN.

where $\alpha(s)$ is the learning rate at $s^{th}$ step, $\alpha_p$ is the peak learning rate and $W$ is the number of warm up steps. The $\alpha_p$ and batch size were tuned for each model individually. We used Google Cloud TPUs with the number of TPUs used shown along with the hyperparameter values in Table 4 We note that we were able to obtain good results with our models using smaller batch sizes but opted for larger batch sizes for faster development iteration speed.