# OpenReview forum: "STUDY: Socially Aware Temporally Causal Decoder Recommender Systems"
_ICLR.cc/2024/Conference — Submitted to ICLR 2024_

### Official Review · Reviewer_54vF · 2023-10-30

**Soundness:** 3 good
**Presentation:** 2 fair
**Contribution:** 2 fair
**Rating:** 5
**Confidence:** 5

**Summary:**

This work focuses on education recommender systems and presents a Socially-aware Temporally caUsal Decoder, named STUDY, to inject social network information while enhancing efficiency.

**Strengths:**

1.	This work, in collaboration with the non-profit organization, aims at enhancing reading engagement among students with reading difficulties, which may promote equity in educational opportunities.
2.	Conducting validation experiments on the dataset from Learning Ally is interesting.
3.	Readable.

**Weaknesses:**

1.	The difference between the social-aware educational recommendation scenario and traditional recommender systems that consider social networks is not obvious or insufficiently discussed.
2.	The advantages of the proposed method are unclear and technical contributions are limited. This work focuses on leveraging social information in Transformer-based sequential recommendation. However, many start-of-art models are capable of this task, while without any causal data leakage issues.
3.	There is less deep analysis or explanation on why such model design can achieve surprising performance.
4.	The baselines for comparison are relatively outdated. The effectiveness of the proposed algorithm should be verified by comparing it with newer baseline algorithms.

**Questions:**

1.	In the educational recommendation studied, is there any significant difference from the general recommendation scenario, causing the existing sociality-aware methods to fail?
2.	Why introduce a separator token in vectors and what is the advantage of doing so?

---

### Official Review · Reviewer_5grQ · 2023-10-30

**Soundness:** 2 fair
**Presentation:** 2 fair
**Contribution:** 2 fair
**Rating:** 3
**Confidence:** 3

**Summary:**

This paper provides a solution for recommending books to students. It concatenates all students‘ interactions in the same class, and uses a causal decoder-only transformer to predict the next book one would interact with.

**Strengths:**

1. Building recommender systems for students (especially for those with dyslexia) is a meaningful and important problem.
2. The proposed method is tailored for this scenario, since students in a same classroom are taught by the same teacher, and have strong peer relationships. The books they would like to read would be very similar.

**Weaknesses:**

1. Novelty is limited. A very simple alternative would be treating a “class” as a “user”, and we recommend the same set of books to all students in this class. I conjecture the performance would be similar.
2. Only one dataset is used for evaluation. Generally we need two or more to validate the effectiveness of the proposed method.
3. Some more recent SOTA recommender systems (e,g, NCF, LightGCN, …) needs to be compared with. Although they haven’t utilized the social information, they are also very strong baselines.

**Questions:**

1. It would be better if the author could provide a more detailed model architcture (e.g. what are the formally defined input, outputs? how many layers of transformers are used? what are the emb size, #heads, …? is the used transformer the vanilla one or variants? …)
2. There are only 2000 books (items). It would be interesting to see ItemPop peformance. (i.e. recommend books with the highest popularity.)
3. If I understand correctly (please correct me if I am wrong), it seems like for every student in the same class, they would be recommended with the same predicted items? Because the model hasn’t model the charateristic for each single user (It just concatenate all interactions of students in the same class)?
If this is the case, the problem could actually be seen as recommending books to a class. We could regard a “class” as a “user”.
4. How many users in the test/validation have not been trained in the train set?
Because SAMN performs so poor in Table 1, meaning that the model might have failed to model user embeddings. If this portion of users is very large, it might not be a fair setting for SAMN.

---

### Official Review · Reviewer_fvs5 · 2023-11-06

**Soundness:** 3 good
**Presentation:** 2 fair
**Contribution:** 3 good
**Rating:** 5
**Confidence:** 4

**Summary:**

This paper introduces STUDY, a socially-aware temporally causal decoder recommender system that takes into account social network information to provide tailored recommendations for demographic groups with unique interests. The system is highly efficient and effective, avoiding complex architectures and circumventing the need for graph neural networks, making it deployable by partners with limited computational bandwidth. The authors compare STUDY to a number of baselines, both sequential and non-sequential, and social and non-social, and show that it outperforms alternative methods in a variety of scenarios.

**Strengths:**

The paper proposes a new architecture and data pipeline for performing socially-aware sequential recommendation. The STUDY architecture is significantly more efficient to learn and train than existing methods and performs joint inference over socially-connected groups in a single forward pass of a modified transformer decoder network. The paper highlights the importance of using social network information effectively in recommender systems, especially for demographic groups with interests that differ from the majority.

The paper proposes a socially-aware temporally causal decoder recommender system (STUDY) that leverages known student hierarchies to improve recommendations. By modeling both within-user and cross-user interactions using a single transformer decoder network, the system is able to effectively incorporate social dynamics into recommendations. This approach has the potential to improve the accuracy and relevance of recommendations, particularly for demographic groups with interests that differ from the majority.

**Weaknesses:**

A potential avenue for enhancing this study might involve elaborating on the parameter optimization process for the compared baselines. The paper could benefit from providing insights on the spectrum of hyperparameters investigated for the baseline models, the approach employed for identifying the optimal hyperparameters, and the influence of different hyperparameters on each model's performance. This level of detail not only enhances the transparency of the research but also provides a better understanding of how each model's performance may vary under different configurations.

An additional enhancement to this study could involve a more exhaustive comparison of the proposed method against a broader array of existing solutions. Such a comprehensive comparative analysis would facilitate a more detailed understanding of the advantages and limitations of the proposed method, and how it measures up against other cutting-edge techniques in the discipline. It is suggested to consider the following recently developed social recommender systems for this comparison:
"Federated Social Recommendation with Graph Neural Network," (TIST).
"SocialLGN: Light Graph Convolution Network for Social Recommendation," (Information Sciences).
"Social Recommendation with Self-Supervised Metagraph Informax Network," (CIKM).

The paper appears to lack explicit details regarding the computational efficiency of the proposed method in comparison to the baseline models. An enhancement to this study could be to furnish more comprehensive information concerning the computational efficiency of the proposed method. This could include aspects such as the time and memory requirements for both training and inference phases, and a comparative analysis of these factors against the baseline models. This additional detail would provide the reader with a clearer understanding of the method's practicality in terms of computational resources.

**Questions:**

The study could be improved by providing detailed information on the parameter optimization process for the compared baselines, including the range of hyperparameters explored, the method for choosing optimal hyperparameters, and the influence of these on each model's performance. Additionally, a more comprehensive comparison of the proposed method with a broader array of existing solutions could be beneficial. The paper currently lacks detailed information on the computational efficiency of the proposed method compared to the established baseline models.

---

### Official Review · Reviewer_Trnv · 2023-11-07

**Soundness:** 3 good
**Presentation:** 2 fair
**Contribution:** 2 fair
**Rating:** 1
**Confidence:** 5

**Summary:**

This paper investigates the social-aware (e.g., in the same classroom) recommendation problem in the audio-book recommendation application. The proposed framework (STUDY) uses a Transformer decoder architecture to solve this problem, with a designated masking strategy to consider social relations (students in the same classroom) and causal relationships (interactions with timestamps). STUDY shows improvement over a private dataset (collected on an educational platform, Learning Ally).

**Strengths:**

S1: Caring about dyslexia or struggling readers is a good point.

S2: the masking strategy to model causal relationships among interactions seems interesting.

**Weaknesses:**

W1: The technical contributions are very limited. The proposed framework (STUDY) is mainly based on a Transformer decoder architecture. The challenges of socially-aware sequential recommendation are not pointed out in the introduction. The drawbacks of existing social recommendation methods are not addressed by the STUDY. The three contributions summarized are not novel.

W2: The evaluation is very weak. There is only ONE social recommendation baseline, SAMN. And there is only ONE recent baseline, SR-GNN. Both methods were proposed in 2019, which are not up-to-date.

W3: Many related works are ignored. The reviewed baselines in Section 3 REVIEW OF BASELINE WORK include one social recommendation model, SAMN. However, it only exploits first-degree social relations (e.g., direct friends). In contrast, the DICER method (Dual Side Deep Context-aware Modulation for Social Recommendation) can model high-order social relations. Furthermore, it is a more recent method proposed in 2021. As a result, the claim in the introduction “comparing the new method to modern recommendation baselines” does not hold fully true.

W4: the newly introduced dataset is not detailed. Since it is a private dataset, will it be released to the public? If not, how can we reproduce the results? What is worse, there is no public dataset evaluated in this paper. There are so many social recommendation datasets, including Epinions, Ciao, et al. It is strongly recommended to report results on benchmarks.

**Questions:**

Q1: The size of the item set is too small, only 2000.
i) what is the practical value of the proposed STUDY model?
ii) for the long-tail items, cold-start items, or rare items, what is the performance of the proposed STUDY model?

Q2: the evaluation metric HITS@N ignores the ranking order of the items. Consider other evaluation metrics like NGCG and MRR to consider the hit position.

Q3: The mathematical notations are messy, unclear, and even not defined. In section 3.2, the notation data D is not defined.

Q4: except Section 4.1, there is no Section 4.2, 4.3... So, why do we need a subsection 4.1?

Q5: check the syntax errors or typos:
“likelihood of the user interacting a with a specific”
“Hence our we redefine the attention operator”
“data relevant to students interaction. with the platform, which falls”

---

### Official Review · Reviewer_GC3n · 2023-11-08

**Soundness:** 3 good
**Presentation:** 2 fair
**Contribution:** 2 fair
**Rating:** 5
**Confidence:** 3

**Summary:**

This paper makes use of the hierarchical social relationships during students to improve the recommender system for education. The authors propose a reasonable method based on causal transformer decoder networks STUDY, for the sake of aggregating the user information in the homogeneous group that avoids introducing temporally disordered recommender history. By doing so, the historic ordered of input data is kept in the temporal decoder. Further experiments demonstrate the effectiveness of STUDY model.

**Strengths:**

1. This paper aims to benefit the recommender system in educational fields, which is of great importance and attracts many investigations in recent.
2. In the methodology section, the authors propose Individual and STUDY, where Individual gives a basic framework to model the recommendation process while STUDY further considers the hierarchical social relationship.
3. Several ablation studies are obvious evidence to support the motivation of STUDY.

**Weaknesses:**

1.	In experiment part, it seems to be strange to only assign IDs to the popular items and discard the remaining unpopular items.
2.	As for the result analysis, the authors do not give a convincing explanation about why the performance of SAMN model under all the three evaluation subset settings are nearly the same, which is difference with the other three models that have diverse performances.
3.	The methodology section is too short and the authors can rearrange the writing structure to enhance the readability.

**Questions:**

see the above comments

---

### Meta-Review · Area_Chair_1rBP · 2023-12-06

**Metareview:**

This paper presents STUDY, a Socially-aware Temporally caUsal Decoder recommender sYstem, to address the audio-book recommendation problem. The paper is well organized. The proposed technique is meaningful as it could potentially benefit students who have dyslexia or are struggling readers. However, reviewers raised major concerns regarding the technical contributions, experiments, paper writing, etc. This paper in its current version is not ready for publication at ICLR.

**Justification For Why Not Higher Score:**

Reviewers raised many valid concerns, but the authors did not provide any responses.

**Justification For Why Not Lower Score:**

N/A

---

### Decision · Program_Chairs · 2024-01-16

Reject